# Application of the American Thyroid Association Risk Assessment in Patients with Differentiated Thyroid Carcinoma in a German Population

**DOI:** 10.3390/biomedicines11030911

**Published:** 2023-03-15

**Authors:** Friederike Eilsberger, Michael C. Kreissl, Christoph Reiners, Adrien Holzgreve, Markus Luster, Andreas Pfestroff

**Affiliations:** 1Department of Nuclear Medicine, University Hospital Marburg, 35043 Marburg, Germany; 2Department of Radiology and Nuclear Medicine, Nuclear Medicine, Division of Nuclear Medicine, University Hospital Magdeburg, 39120 Magdeburg, Germany; 3Department of Nuclear Medicine, University Hospital Würzburg, 97080 Würzburg, Germany; 4Department of Nuclear Medicine, University Hospital, LMU Munich, 81377 München, Germany

**Keywords:** differentiated thyroid cancer, American Thyroid Association, German population

## Abstract

Background: The American Thyroid Association (ATA) uses criteria to assess the risk for persistent disease in differentiated thyroid carcinoma (DTC) after radioiodine therapy (RAI). There are no data available showing that this classification can be adopted unadjusted by Germany. Aim: The aim of our study is to investigate whether the ATA classification can be applied to a German population for short-term prognosis. Furthermore, we investigated the influence of an age cutoff value. Methods: We retrospectively analyzed 121 patients who were referred to our tertiary referral center. Patients were classified into risk categories, and the therapy response was determined according to ATA. Results: A total of 73/83 (88%) ATA low-risk patients and 12/19 (63%) intermediate-risk patients showed an excellent response; 2/19 (11%) high-risk patients had a biochemical, and 6 (31%) had a structural incomplete response. Of all 39 patients ≥55 years, 84% had an excellent response. Using a cut off of 50 years, 50/62 (81%) of the older patients showed an excellent response. Conclusion: The ATA risk classification is able to estimate the response to RAI therapy in a German population. A shift from 55 to 50 years as an age cutoff value does not result in any relevant change in the treatment response.

## 1. Introduction

The American Thyroid Association (ATA) 2015 Guideline uses several criteria to assess the risk for persistent or recurrent disease in differentiated thyroid carcinoma (DTC) after radioiodine therapy (RAI), resulting in a classification of low-risk, intermediate-risk, and high-risk groups [1] (Table 1). 

In addition, response to RAI can be categorized based on certain characteristics (Table 2). Based on these groups’ characteristics, the risk for persistent or recurrent disease can be reassessed during the course of the disease.

Currently, there are efforts to transfer the classification system of the ATA to Europe. The European Thyroid Association (ETA) has adopted the classification of risk groups in their last consensus statement, without further adjustment [2]. However, the longstanding iodine deficiency in Germany is followed by a high prevalence of multinodular goiter; thus, the general approach of the U.S. cannot be transferred without further assessment [3,4]. Recently, Maneck et al. showed that less than 15.2 % of surgeries in Germany revealed a malignancy [5]. Comparing these results with the data from the Surveillance, Epidemiology, and End Results (SEER) database, we can see that about 36 % of thyroid surgeries in the U.S. result in a DTC diagnosis, a frequency more than twice that of Germany. Likewise, the clinical picture in Germany is characterized by limited use of preoperative diagnostics (e.g., fine needle biopsy) [6,7]. The attention in the U.S. is presumably more focused on malignancies in structural thyroid abnormalities, likely resulting in more frequent operations with oncologic indication, while DTC in Germany is often discovered incidentally.

Because of these structural differences, the aim of our study is to investigate whether the ATA classification can be applied to a German population for short-term prognosis and prediction of disease persistence after first RAI.

As a second aim, we investigated the impact of an age cutoff value as parameter in our population. The impact of this cutoff is of additional interest since, in some studies, older age is considered an influencing factor for poorer response to therapy, while other studies cannot find a correlation [8,9].

## 2. Materials and Methods

### 2.1. Patients

We retrospectively examined the data of 121 patients who were referred to our tertiary referral center for the first course of ^131^I therapy between 1 July 2017 and 30 June 2019 and were followed until 31 October 2020. Median age at diagnosis was 51 years, and 85/121 (70%) patients were women. The most common histopathologic type was papillary thyroid cancer in 108/121 (89%) of the patients. Basic characteristics can be found in Table 3. All other information on the study population is listed in Table 4.

Patients received activities ranging from 2 to 11 GBq of ^131^I (median 3.7 GBq), depending on the known tumor stage (such as lymph node and distant metastases) at this point in time. Therapy was performed under exogenous thyrotropin stimulation with a time interval of one to seven months (median one month) after complete thyroidectomy.

After RAI, patients were grouped into low-risk, intermediate-risk, and high-risk ATA groups by experienced nuclear medicine physicians based on their histopathologic findings, tumor characteristics, and post-therapeutic ^131^I scan.

All patients were followed up with thyroglobulin (Tg) measurements. Additionally, 112 patients underwent cervical ultrasound 2–32 months (median 6 months) after therapy in our department. Cervical ultrasound revealed thyroid remnant in nine patients, a suspicious thyroid bed lesion in two patients, and suspicious lymph node(s) in five patients.

Furthermore, 92 patients received imaging with ^131^I diagnostic scintigraphy 5–23 months (median 11 months) after therapy. A total of 15 patients showed an uptake in the thyroid bed, and a suspicious lymph node was found in 1 patient. One patient received inconspicuous ^123^I diagnostic scan 17 months after RAI.

A total of 16 patients received further diagnostics with ^18^F-FDG-PET/CT imaging 2–24 months (median 7 months) after therapy, 6 of whom also underwent ^131^I diagnostic scintigraphy.

Moreover, the response to therapy was classified using the information obtained during follow-up (Tg value, ultrasound,^131^I diagnostics, second RAI, and further imaging as excellent, indeterminate, biochemically incomplete, and structurally incomplete, according to the ATA classification.

### 2.2. Data Extraction

From our clinical files, we extracted data on histopathological findings, tumor characteristics, and imaging (e.g., cervical ultrasound, post-RAI scans, and ^131^I diagnostic scintigraphy). If a procedure was not explicitly reported (e.g., ^131^I diagnostic imaging), it was assumed to have not been performed. This study was approved by the local ethics committee (Az.:ek_mr_21072021_eilsberger). For the present study, data and results were employed as recorded in the attending physicians’ reports. Pathology data were employed as recorded in the pathologists’ written report. All patients were evaluated according to the 8th edition of the TNM stage [10].

### 2.3. Analysis

Data are reported descriptively either as numbers or percentage values. Microsoft^®^ Excel version 16.68 (Washington, DC, USA) was used for data collection and processing.

## 3. Results

### 3.1. Risk Classification into the Different ATA Groups

In total, 83/121 (69%) patients were classified to be in the low-risk group, and 19/121 (16%) each in the intermediate-risk and high-risk groups.

In total, 73/83 (88%) ATA low-risk patients had an excellent response, 5 (6%) had an indeterminate response, 3 (4%) had a biochemical response, and 2 (2%) had a structural incomplete response. Regarding the intermediate risk patients, 12/19 (63%) showed an excellent response, 2 (11%) showed an indeterminate response, 1 (5%) showed a biochemical response, and 4 (21%) showed a structural incomplete response. Of the 19 high risk patients, 9 (47%) had an excellent response, 2 (11%) had an indeterminate response, 2 (11%) had a biochemical response, and 6 (31%) had a structural incomplete response. Summarizing all patients of the different risk groups, we found that 94/121 (78%) patients had an excellent response, 9/121 (7%) had an indeterminant response, 6/121 (5%) had a biochemical incomplete response, and 12/121 (10%) patients had a structural incomplete response. These results can be found in Table 5.

### 3.2. Patients Age as a Factor Concerning Response in Papillary Thyroid Carcinoma

A total of 108 patients had the histopathological diagnosis of a papillary thyroid carcinoma. Since this represents the main group in our population, we evaluated these patients again separately according to age. We used a cutoff value of 50 years, as suggested by Van Velsen et al., and with 55 years being the established cutoff in the current TNM system (8th edition).

We applied a cut off value 55 years, and 53/69 (77%) patients younger than 55 years had an excellent response. In the group of older patients, 84% had an excellent response. Using a cut off of 50 years, 36 of the 46 (78%) patients younger than 50 years of age had an excellent response. In the 50 years or older group, 50/62 (81%) patients had an excellent response. The groups differentiated by the age cutoff did not differ in their assignment to the ATA risk groups. Further information on the age cutoff can be found in Table 6.

## 4. Discussion

Comparing our results in terms of therapy response in different risk groups to those results referred to in the ATA guideline from the literature, we see several similarities. An excellent response to initial RAI is described in the ATA guideline in 86–91% of low-risk patients, to a lesser extent in 57–63% of intermediate-risk patients, and in at least 14–16% of high-risk patients [11,12,13,14]. In our population, we found an excellent response in 88% of low-risk patients, 63% of intermediate-risk patients, and 47% of high-risk patients. Therefore, our results in the low-risk and intermediate-risk groups are very comparable, with the results of our population in the high-risk group being distinctly better.

Biochemical incomplete response was detected according to Tuttle et al., Vaisman et al., and Pitoia et al. in 10–15% of low-risk patients, in 14–22% of intermediate-risk patients, and in 12–14% of high-risk patients. In addition, the authors found structural incomplete response in 2–7% of low-risk patients, in 21–34% of intermediate-risk patients, and in 56–72% of high-risk patients [11,12,14].

Compared with the literature, we found fewer biochemical incomplete responses in our population, with a distribution of 4% in the low-risk group, 5% in the intermediate-risk group, and 10.5% in the high-risk group. Our population showed a similar number of structural incomplete responses in the low-risk and intermediate-risk groups, with 2% and 21% of patients affected, respectively. However, 32% of patients were affected in the high-risk group. This is compatible with the high number of excellent responses in this group.

The classification of the American Thyroid Association is confirmed with regard to tumor persistence. About 10–15% of patients in the low-risk group were affected, even if our population shows a slightly lower level with 6% of low-risk patients affected. The fact that even patients with a low risk can show persistent disease, even if they are few, should be kept in mind. This is an important aspect to consider in everyday clinical practice and should be carefully taken into account, especially with regard to the currently discussed indication for radioiodine therapy [15,16]. A recent German position paper by nuclear medicine physicians and endocrine surgeons, written by Schmidt et al., summarizes areas of disagreement based on a literature review in comparison with the ATA [15]. The ATA and the ETA are cautious about RAI in low-risk patients and see the indication for it only in individual exceptional cases. However, individual countries such as Germany and associations such as the Society of Nuclear Medicine and Molecular Imaging (SNMMI) and the European Association of Nuclear Medicine (EANM) see a possible advantage for low-risk patients [16]. Additionally, many patients (12–50%, according to Perros et al. [17]) show lymph node metastases, even in pT1a tumors. As such, the percentage of Nx-operated patients is high, especially in Germany, which is an iodine-deficient country (in our study, 40/121 (33%) patients received Nx thyroidectomy, which does not allow for any statement about possibly present lymph node metastases). Radioiodine therapy can be an adjuvant approach in these supposedly low-risk patients [16]. Accounting for the well-known high incidence of lymph node metastases, adjuvant therapy, as opposed to remnant ablation, is therefore traditionally pursued in Germany. This is why patients usually receive activities that are considered to be therapeutically appropriate [15,16,18]. Tuttle et al. emphasize in their joint statement by the ATA, EANM, SNMMI and ETA that a clear distinction should be made between a remnant ablation and an adjuvant therapy concept, specifically that the latter is heavily influenced by local conditions [19]. Deciding to pursue an adjuvant therapy concept with the administration of higher activities (even if they are still in the fairly low range) may increase the risk of side effects.

The fact that other European countries also deviate from the recommendations of the ATA with regard to RAI is demonstrated by Lamartina et al. In their paper, the authors state that in Italy (also long-time iodine deficiency country), RAI is frequently used for multifocal micro PTCs, which is in clear contrast to the 2009 ATA guideline [20].

In a current analysis of the ESTIMABL2-trial patients with low-risk thyroid cancer after thyroidectomy, a follow-up strategy that did not involve the use of radioiodine (1.1 GBq) was non-inferior to an ablation strategy with radioiodine, regarding the occurrence of functional, structural, and biologic events at 3 years’ follow-up [21]. The main criticisms of this study, as stated by specialists such as those from the group of Tuncel et al., are the low activity of 1.1 GBq (which induces remnant ablation, no adjuvant therapy [16]), the study population (pT1a/b, N0/x) covering only a part of the low-risk group, and the short follow-up period, which, according to the work of Sawka et al., should be at least 10 years [22,23].

The good response to initial RAI in the high-risk group, more common in our German population than in the literature, may have various causes. It is possible that patients with previously unknown metastases (assigned to the high-risk group after post-therapeutic RAI imaging) received higher activities of ^131^I compared with patients in the U.S., where RAI (with accompanying post-therapeutic ^131^I whole-body scintigraphy) is not used as intensively as in Germany. If no RAI is performed, metastases may remain undetected (and at that moment also untreated), at least for a certain time. This may lead to a bias—advanced disease detected (and treated) early on with a higher tumor classification, compared with detection of clinically evident advanced disease initially (without the use of RAI), leading to a worse outcome.

Another widely discussed aspect is the influence of age on the response to therapy. While Shah and Alzahrani were able to demonstrate an association between treatment response and age (cut off Shah et al.: 55 years), other authors such as Campennì et al. were unable to demonstrate this relationship [8,9,24].

We analyzed the influence of age on therapy response in the papillary thyroid cancer subgroup using age cutoffs of 50 years and 55 years. We used an age cutoff of 50 years, as suggested by Van Velsen et al., to predict disease-specific survival (DSS) in European DTC patients as well as in the papillary thyroid cancer subgroup. We also used an age cutoff of 55 years, as it is the currently established cutoff value of the AJCC/UICC staging system 8th edition [25]. Due to the small number of cases, we did not consider it to be conclusive to study other subgroups.

Similar to Campennì et al., in our population, a correlation between age at first diagnosis and treatment response was not detectable for the papillary thyroid cancer patients.

Our study has certain limitations. The most important limitation is the small number of patients included. This especially holds true for the advanced cases, but also for the non-papillary subtypes of thyroid cancer. The limited number of cases may result in statistical imbalances and failure to detect correlations due to of lack of statistical power. We performed an analysis in a single center. Furthermore, our single center is a tertiary care center with high-volume referring surgeons, which can also lead to misrepresentation. A multi-center analysis could improve the study results. Although the treatment protocol in our center falls well within the bounds of various international guidelines, a single-center analysis may not reflect standard clinical practice in other institutions. In addition, the study evaluated data retrospectively and did not collect data prospectively.

## 5. Conclusions

The ATA risk classification in this short-term response study appears to be able to estimate the response to RAI therapy in the German patient population. A shift from 55 to 50 years as an age cutoff value does not lead to any relevant change in the treatment response results.

## Figures and Tables

**Table 1 biomedicines-11-00911-t001:** Risk of recurrence according to the ATA guideline [1].

**Low risk**	– papillary thyroid cancer – no local or distant metastases – no remaining macroscopic tumor tissue – no invasion in loco-regional tissues or structures – no aggressive histology (e.g., tall cell, hobnail variant, columnar cell carcinoma) – if ^131^I is given: no radioiodine (RAI)-avid metastatic foci outside the thyroid bed – no vascular invasion – clinical N0 or ≤5 N1 micrometastases (<0.2 cm) – intrathyroidal, encapsulated follicular variant of papillary thyroid cancer – intrathyroidal, well-differentiated follicular thyroid cancer with capsular invasion and no or minimal (<4 foci) vascular invasion – intrathyroidal, papillary microcarcinoma, unifocal or multifocal, including BRAFV600E mutated (if known)
**Intermediate risk**	– microscopic invasion of tumor into the perithyroidal soft tissues – RAI-avid metastatic foci in the neck – aggressive histology (e.g., tall cell, hobnail variant, columnar cell carcinoma) – papillary thyroid cancer with vascular invasion – clinical N1 or >5 pathologic N1 (<3 cm) – multifocal papillary microcarcinoma with extra thyroidal extension and BRAFV600E mutated (if known)
**High risk**	– macroscopic invasion of tumor into the perithyroidal soft tissues – incomplete tumor resection – distant metastases – postoperative serum thyroglobulin suggestive of distant metastases – pathologic N1 with any metastatic lymph node ≥3 cm – follicular thyroid cancer with extensive vascular invasion (>4 foci)

**Table 2 biomedicines-11-00911-t002:** Therapy response according to the ATA guideline [1].

**Excellent response**	– negative imaging – suppressed Tg < 0.2 ng/mL or stimulated Tg < 1 ng/mL	– 1–4% recurrence – <1% disease specific death
**Indeterminate response**	– nonspecific findings on imaging studies – faint uptake in thyroid bed on therapy scan – nonstimulated Tg < 1 ng/mL – stimulated Tg < 10 ng/mL or anti-Tg antibodies stable/declining	– 15–20% will have structural disease in follow-up – <1% disease-specific death
**Biochemical incomplete response**	– negative imaging – suppressed Tg ≥ 1 ng/mL or stimulated Tg ≥ 10 ng/mL or rising anti-Tg antibody levels	– 30% spontaneously evolve to no evidence of disease (NED) – 20% achieve NED after additional therapy – 20% develop structural disease – <1% disease specific death
**Structural incomplete response**	structural or functional evidence of disease – with any Tg level – with or without anti-Tg antibodies	50–85% continue to have persistent disease despite additional therapy – disease-specific death rates: 11% with loco-regional metastases 50% with structural distant metastases

**Table 3 biomedicines-11-00911-t003:** Basic characteristics study population.

	N = 121	
Age at diagnosis in years	Median 51 (range, 14–86)	
Sex	Female Male	85 36
Histology	Papillary Follicular Others	108 7 6

**Table 4 biomedicines-11-00911-t004:** Therapy and follow-up study population.

	*n* = 121	
TNM before RAI (8th edition)	T1 T2 T3 T4	61 40 18 2
	N0/x N1	37/40 44
	M1	3
AJCC/UICC classification stage	I II III Iva Ivb	103 14 2 0 2
Initial ATA classification	Low risk Intermediate risk High risk	83 19 19
Radioiodine therapy	Activity (in MBq)	3.700 (2.000–11.000)
	Interval to surgery in months	1 (1–7)
	Thyroglobulin (not stimulated) (ng/mL) *	1 (<0.04–20.626)
	Thyroglobulin (stimulated) (ng/mL) *	7.3 (<0.04–47.262)
Follow-up	Thyroglobulin (not stimulated) (ng/mL) *	<0.04 (<0.04–4046)
	Positive thyroglobulin antibodies	6 patients
	Cervical ultrasound	112 patients
		Median 6 months (2–32) Thyroid remnant: 9 patients Suspicious thyroid bed lesion: 2 patients Suspicious lymph node: 5 patients
	^131^I diagnostic scintigraphy	92 patients
		Median 11 months (5–23) Uptake in thyroid bed: 15 patients Suspicious lymph node: 1 patient Thyroglobulin (stimulated) (ng/mL) * <0.04 (<0.04–4046)
	^123^I diagnostic scintigraphy	1 patient; not suspicious, 17 months after RAI
	Second RAI	9 patients
		Median 9 months (6–16)
	^18^F-FDG-PET/CT	16 patients
		Median 7 months (2–24) in 6 patients additional to 131I diagnostic scintigraphy

* Thyroglobulin (Tg) detection threshold 0.04 ng/mL.

**Table 5 biomedicines-11-00911-t005:** Response in ATA risk stages.

	Excellent Response	Indeterminate Response	Biochemical Incomplete Response	Structural Incomplete Response	Total
Low risk	73/83 (88%)	5/83 (6%)	3/83 (4%)	2/83 (2%)	83
Intermediate risk	12/19 (63%)	2/19 (11%)	1/19 (5%)	4/19 (21%)	19
High risk	9/19 (47%)	2/19 (10.5%)	2/19 (10.5%)	6/19 (32%)	19
	94	9	6	12	121

**Table 6 biomedicines-11-00911-t006:** Therapy response of papillary thyroid cancer patients in different age cutoff groups and ATA categories.

Population N = 108	Different Age Cutoff Groups and ATA Risk Categories			
	<55 Years N = 69/108 (63%)	>55 Years N = 39/108 (37%)	<50 Years N = 46/108 (44%)	>50 Years N = 62/108 (56%)
Excellent response	53/69 (77%)	33/39 (84%)	36/46 (78%)	50/62 (81%)
	41/53 (77%) low risk 8/53 (15%) int. risk 4/53 (8%) high risk	28/33 (85%) low risk 1/33 (3%) int. risk 4/33 (12%) high risk	26/36 (72%) low risk 8/36 (22%) int. risk 2/36 (6%) high risk	43/50 (86%) low risk 1/50 (2%) int. risk 6/50 (12%) high risk
Indeterminate response	6/69 (9%)	3/39 (8%)	4/46 (9%)	5/62 (8%)
	4/6 (67%) low risk 2/6 (33%) int. risk 0 high risk	1/3 (33%) low risk 0 int. risk2/3 (67%) high risk	3/4 (75%) low risk 1/4 (25%) int. risk 0 high risk	2/5 (40%) low risk 1/5 (20%) int. risk 2/5 (40%) high risk
Biochemical incomplete response	6/69 (9%)	0 (0%)	4/46 (9%)	2/62 (3%)
	3/6 (50%) low risk 1/6 (17%) int. risk 2(6 (33%) high risk		1/4 (25%) low risk 1/4 (25%) int. risk 2/4 (50%) high risk	2/2 (100%) low risk
Structural incomplete response	4/69 (5%)	3/39 (8%)	2/46 (4%)	5/62 (8%)
	0 low risk 1/4 (25%) int. risk 3/4 (75%) high risk	1/3 (33.3%) low risk 1/3 (33.3%) int. risk 1/3 (33.3%) high risk	0 low risk 0 int. risk 2/2 (100%) high risk	1/5 (20%) low risk 2/5 (40%) int. risk 2/5 (40%) high risk

Low risk	48/108 (45%)	30/108 (28%)	30/108 (28%)	48/108 (45%)
Int. risk	12/108 (11%)	2/108 (2%)	10/108 (9%)	4/108 (4%)
High risk	9/108 (8%)	7/108 (6%)	6/108 (5%)	10/108 (9%)

Abbreviation: Int. = intermediate.

## Data Availability

The data that support the findings of this study are available from the corresponding author (F.E.) upon reasonable request.

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
