# Peer review of "Application of the American Thyroid Association Risk Assessment in Patients with Differentiated Thyroid Carcinoma in a German Population"

_biomedicines, 2023, doi:10.3390/biomedicines11030911_

Round 1
Reviewer 1 Report
Thank the Editor to give me the opportunity to revise this article.
The manuscript is of great interest in the field of current research. The work is well written and adequately structured.
Having said that, I have to highlight some very important caveats to the authors:
1) in the Introduction section please rephrase the sentence in lines 36-37 in a more understandable way.
2) table 1, point 7: to be more precise, the “low risk” category of the “ATA risk stratification system” states that “if 131I is given… no radioiodine avid metastatic foci […]”. For this reason I suggest authors to modify this sentence since it gives different meaning to all the purpose of the risk stratification (ie just few patients belonging to this category usually undergo to preoperative 131I scan)
3) The authors present data collected retrospectively from 121 patient (all) treated with 131I between 2017 and 2019. 83/121 of them have been classified as “ATA low risk” category: the ATA guidelines on the management of DTC, published in 2016 (before the time interval considered by the current paper), recommends the use of 131I ablation for “low risk” patients only in those few with worrisome features (was it the case of these 83 patients?). Moreover, the median dose delivered in the study was 3,7 GBq (ie 100 mCi), while both the ATA guidelines and the 2022 ETA consensus recommend for this type of patients a much lower dose, ie 1,1 GBq (30 mCi) for ablation purpose (103/121 patients have been staged “I”, so presumably most of them was without extensive distant disease at diagnosis as then demonstrated during the follow-up; so the questions are: did they really need the 131I ablation?at even higher dose than recommended?).
Furthermore, the 2016 ATA guidelines highlighted the need of prospective and controlled data when discussing the available evidences for the recommendation of 131I therapy. Some of the new data are finally accessible (see ESTIMABL2 trial, published in March 2022 and not cited by the authors), and they confirmed that “patients with low-risk thyroid cancer undergoing thyroidectomy, a follow-up strategy that did not involve the use of radioiodine (at 1.1 GBq dose) was noninferior to an ablation strategy with radioiodine regarding the occurrence of functional, structural, and biologic events at 3 years”.
Therefore, based on the current guidelines (ATA 2016 and 2022 ETA consensus) and recent high-quality evidences (ESTIMABL2 trial), the vast majority of the patients of the current study have been probably overtreated and this aspect could have really impacted on the subsequent results obtained during the follow-up (ie “the response to therapy reclassification”).
4) As stated by authors in section “2.3 Analysis”, no statistics have been carried out. Therefore, I consider the term “significantly” inappropriate in a such context, as it is done in line 150.
Overall, this study is based on a valid purpose. However, the main objective (i.e. to compare the applicability of the “ATA risk classification system and the response to therapy reclassification” on a German population sample), in my opinion, is largely not achieved as the guidelines themselves have not been considered in the correct way.
Reviewer 2 Report
The idea of the article is novel and essential, since it is the only management workflow we follow for treatment.
Going through the abstract with the values of correct matching risk stratification with outcomes (73+12+8)/121 thus 77% only were correctly allocated to the corresponding risk tier.
Introduction: Glad that German and non-German values were compared in the introduction.
ATA classification followed was 2009 or 2015?
Section 2.3 requires details on the software used for analysis and types of tests performed.
However, the conclusion does not reflect the study design and analysis.
Round 2
Reviewer 1 Report
I thank the authors since all points of criticism have been sufficiently addressed in the discussion and explained.
However, I would add some points still:
1. I suggest them to add in the discussion a brief comment and comparison with the result of the paper "Are Evidence-Based Guidelines Reflectedin Clinical Practice? An Analysis of ProspectivelyCollected Data of the Italian Thyroid Cancer Observatory", Lamartina L et al, Thyroid 2017. In this study, the authors aimed to explore if the management of thyroid cancer in Italy differ from what suggested by the ATA guidelines. For example, it emerged a frequent use of RAI for microPTCs that were multifocal (44% vs. 16.8% of unifocal microPTCs) which was in clear contrast with the 2009 ATA guideline, even if the median dose used was lower (50 mCi) than those administered in intermediate- and high-risk cases(100 mCi), consistently with 2009 ATA recommendations. Moreover, as a interesting point of similiraty with the current manuscript, Italy is an european country that reached the iodine-sussificency status only in 2021.
2. "With the known high incidence of lymph node involvement, the concept of adjuvant therapy is therefore traditionally pursued in Germany, which is why patients receive higher activities that are supposed to be therapeutically appropriate (we have added relevant literature evidence in the discussion). We added this point in detail to our discussion and we agree with the reviewer that probably a percentage of our patients undergo overtreatment following this established concept in Germany"
Please consider in the discussion that this approach carries an higher rate of side effects associated with high 131I activities
Author Response
Thank you very much for your effort and time. We have gladly taken up your suggestions and implemented them in the paper.
1) 1. I suggest them to add in the discussion a brief comment and comparison with the result of the paper "Are Evidence-Based Guidelines Reflected in Clinical Practice? An Analysis of Prospectively Collected Data of the Italian Thyroid Cancer Observatory", Lamartina L et al, Thyroid 2017. In this study, the authors aimed to explore if the management of thyroid cancer in Italy differ from what suggested by the ATA guidelines. For example, it emerged a frequent use of RAI for microPTCs that were multifocal (44% vs. 16.8% of unifocal microPTCs) which was in clear contrast with the 2009 ATA guideline, even if the median dose used was lower (50 mCi) than those administered in intermediate- and high-risk cases (100 mCi), consistently with 2009 ATA recommendations. Moreover, as a interesting point of similiraty with the current manuscript, Italy is an european country that reached the iodine-sussificency status only in 2021.
We thank the reviewer for contributing this interesting work by Lamartina et al. and have added it to the discussion.
- "With the known high incidence of lymph node involvement, the concept of adjuvant therapy is therefore traditionally pursued in Germany, which is why patients receive higher activities that are supposed to be therapeutically appropriate (we have added relevant literature evidence in the discussion). We added this point in detail to our discussion and we agree with the reviewer that probably a percentage of our patients undergo overtreatment following this established concept in Germany"
Please consider in the discussion that this approach carries an higher rate of side effects associated with high 131I activities
We thank the reviewer for this point and added it to the discussion.
Reviewer 2 Report
Thanks for the respond.
All inquiries were addressed.
Author Response
We thank the reviewer for his time and effort and are happy to have answered all points.
Round 3
Reviewer 1 Report
Accept in the present form